# Validation of an HPLC-CAD Method for Determination of Lipid Content in LNP-Encapsulated COVID-19 mRNA Vaccines

**DOI:** 10.3390/vaccines11050937

**Published:** 2023-05-04

**Authors:** Xiaojuan Yu, Chuanfei Yu, Xiaohong Wu, Yu Cui, Xiaoda Liu, Yan Jin, Yuhua Li, Lan Wang

**Affiliations:** 1Key Laboratory of the Ministry of Health for Research on Quality and Standardization of Biotech Products, National Institutes for Food and Drug Control, Beijing 102629, China; yuxiaojuan@nifdc.org.cn (X.Y.); yuchuanfei@nifdc.org.cn (C.Y.);; 2Thermo Fisher Scientific (China) Co., Ltd., Shanghai 201206, China

**Keywords:** mRNA vaccine, charged aerosol detector, lipid nanoparticles, method validation

## Abstract

Lipid nanoparticles (LNPs) are widely used as delivery systems for mRNA vaccines. The stability and bilayer fluidity of LNPs are determined by the properties and contents of the various lipids used in the formulation system, and the delivery efficiency of LNPs largely depends on the lipid composition. For the quality control of such vaccines, here we developed and validated an HPLC-CAD method to identify and determine the contents of four lipids in an LNP-encapsulated COVID-19 mRNA vaccine to support lipid analysis for the development of new drugs and vaccines.

## 1. Introduction

SARS-CoV-2 was initially identified as the pathogen that caused the outbreak of pneumonia cases in late December 2019 and spread quickly across the world [1,2,3]. The spread of this pathogen was declared a global health emergency by the World Health Organization, and the disease caused by SARS-CoV-2 was named coronavirus disease 2019 (COVID-19). SARS-CoV-2 is a coronavirus similar to SARS-CoV, but its spread is 40 times higher than that of SARS-CoV [4]. The COVID-19 pandemic poses an unprecedented threat to human health. With its continuous mutation and evolution, SARS-CoV-2 has had a massive negative impact on the global economy and society. Vaccines are an important means to prevent infectious diseases and protect public health [5,6,7]. To meet the need for a large number of effective vaccines, messenger ribonucleic acid (mRNA) vaccines have become a primary focus of research in the pharmaceutical industry and in the field of biotechnology [8,9]. Compared with other vaccine platforms—such as inactivated or attenuated virus vaccines, genetically engineered recombinant vaccines, and viral-vector-based vaccines—mRNA vaccines are characterized by transient expression, dual mechanisms of humoral and cellular immunity, a simple production process, high production efficiency, and low cost; thus, they have outstanding advantages in dealing with large-scale and emerging epidemics. However, mRNA vaccines also have some disadvantages, such as their instability, high immunogenicity, and low level of delivery efficiency [10]. The development of tailored delivery systems can further improve the stability and transfection ability of mRNA vaccines. Lipid nanoparticles (LNPs) are currently the most widely used delivery system for these vaccines [11,12,13,14].

Here, we studied an LNP system for COVID-19 mRNA vaccine delivery (shown in Figure 1) using the following four types of lipids: a cationic lipid with an amine functional group that interacts with mRNA via ionic interactions (9001), a PEG2K (polyethylene glycol)-lipid conjugate (1,2-dimyristoyl-rac-glycero-3-methylpolyoxyethylene, PEG2K-DMG), cholesterol (CHOLE), and a zwitterionic helper phospholipid (1,2-distearoyl-sn-glycero-3-phosphocholine, DSPC). Lipids are prone to aggregation and degradation during storage and transportation, which affects the efficiency and safety of COVID-19 mRNA vaccines [15]. Therefore, the contents of individual lipids are critical for ensuring the quality of COVID-19 mRNA vaccines, and they need to be strictly controlled during the production and release stages.

High-performance liquid chromatography (HPLC) is a commonly used modern analytical technique in pharmaceutical analysis. It is widely used in drug analysis due to its high efficiency, speed, sensitivity, and automation. Multiple different detectors can be used in conjunction with HPLC to meet the analysis requirements of different substances, including ultraviolet detectors (UV), refractive index detectors (RID), evaporative light scattering detectors (ELSD), mass spectrometers (MS), and pulse amperometric detectors (PAD).

Charged aerosol detection (CAD) is a new type of universal detector that has been developed in the past decade. Its characteristics include compatibility with gradient elution, response values that do not depend on the structural properties and ionization efficiency of the substance, and the ability to detect non-volatile and semi-volatile substances. When the CAD detector is used in conjunction with HPLC, the HPLC eluent is atomized by nitrogen collision in the CAD detector’s spray chamber. Smaller droplets containing the analyte dry at room temperature to form solute particles. The particle surfaces are positively charged by collision with charged nitrogen. The ion-trap device with low negative voltage removes excess charged nitrogen with high migration rates, and the charged analyte particles with low migration rates transfer their charges to a particle collector. Finally, the electric charge of the analyte is measured and converted into an electrical signal by a highly sensitive electrostatic detection device. The intensity of the signal is proportional to the mass of the solute [16,17,18].

Liquid chromatography with CAD detection for drug analysis has been included in the pharmacopoeias of various countries. For example, both the British Pharmacopoeia 2021 (BP 2021) and the United States Pharmacopeia 2021 (USP 2021) use CAD detectors to investigate substances related to the contrast agent gadobutrol. The determination of deoxycholic acid content in USP43/NF38 also uses CAD detectors. The current version of the Chinese Pharmacopoeia 2020 mentions the CAD detector in the general HPLC method and provides a brief introduction.

Lipids do not contain chromophores and do not absorb ultraviolet radiation under non-derivatization conditions. They are usually difficult to volatilize. Therefore, ELSD detectors and CAD detectors are more suitable for lipid content detection [19]. The principles of the two methods are similar in the early stage of atomization, but different detection methods are used. The amount of light scattered by ELSD is related to the concentration of lipids, while CAD measures the charge carried by the analyte particles and correlates it with mass of analyte. CAD is more sensitive and can detect a wider range of lipid content [15,20,21,22].

We developed a robust and efficient HPLC-charged aerosol detection method (HPLC-CAD) for lipid analysis of an LNP-encapsulated COVID-19 mRNA vaccine and performed a comprehensive validation of the method. The method is stable, efficient, and sensitive, and can be used to determine the contents of four kinds of lipids in COVID-19 mRNA vaccines. Though some of the lipids mentioned here are proprietary and cannot be disclosed, it is still valuable to highlight this methodology in the growing field of COVID-19 mRNA vaccine technology.

## 2. Materials and Methods

### 2.1. Chemicals and Reagents

Methanol (liquid chromatograph ion spray mass spectroscopy/LCMS-grade), ethyl alcohol (LCMS-grade), and triethylamine acetate (high-performance liquid chromatography/HPLC-grade) were purchased from Thermo Fisher Scientific (Waltham, MA, USA); 1,2-dimyristoyl-rac-glycero-3-methoxypolyethylene glycol-2000 (DMG-PEG2K) was purchased from NOF America Corporation (Irvine, CA, USA); 1,2-distearoyl-sn-glycero-3-phosphocholine (DSPC) (>99%) and cholesterol were purchased from Nippon Fine Chemical (Osaka, Japan); and 9001 was synthesized by Abogen (Suzhou, China). Monobasic potassium phosphate, disodium hydrogen phosphate, and sodium chloride were purchased from Sigma-Aldrich (St. Louis, MO, USA). Water (AqDD: <0.55 μS/cm) was dispensed using a Milli-Q filter system from Millipore (Burlington, MA, USA). The COVID-19 mRNA vaccine samples were maintained in our laboratory.

### 2.2. HPLC-CAD Conditions

Thermo Scientific UltiMate 3000 HPLC system consisting of a vacuum degasser, quaternary pump, autosampler, thermostated column compartment, and charged aerosol detector was used to develop the method. Pure nitrogen from nitrogen generator was used at a preset manufacture pressure of 60.7 psi. Samples were injected using a fixed-loop injection at 10 μL with a 100 μL sample loop. COVID-19 mRNA vaccine samples were separated using a Waters XBridge Peptide BEH C18 Column (300 Å pore size, 3.5 µm particle size, 4.6 mm ID × 150 mm column; Waters Corp, Wilmington, MA, USA). The detection temperature was set at 50 °C, the column temperature at 55 °C, and the flow rate at 1.0 mL/min. The injection volume was 10 μL. Data were processed using Thermo Scientific^TM^ Dionex^TM^ Chromeleon^TM^ 7 Chromatography Data System Version 7.3 software.

### 2.3. Mobile Phase

The mobile phase comprised two eluents in a gradient elution mode. Mobile phase A was 0.01 M triethylamine acetate in double distilled water, while mobile phase B was an organic eluent containing 0.01 M triethylamine acetate in ethyl alcohol. The first step of the gradient began from 80% B to 100% B in the first 5 min, followed by holding at 100% B until 13 min; at 13.1 min, the %B was decreased to the initial condition (80%) and equilibrated for 4.0 min before the next sample injection.

### 2.4. Solution and Sample Preparation

#### 2.4.1. Standard Solution and Quality Control Solution

First, DMG-PEG2K, DSCP, CHOLE, and 9001 were individually dissolved in pure methanol to prepare stock standard solutions of 1 mg/mL, 2 mg/mL, 4 mg/mL, and 12 mg/mL, respectively. Then, the mother solution of the standard was prepared by transferring 1 mL of each of the four lipid stock standard solutions into a 10 mL volumetric flask, diluting with pure methanol, and fixing the volume to the scale. Finally, 50 μL, 75 μL, 100 μL, 125 μL, 150 μL, and 200 μL of the stock solution were added to the appropriate volume of mobile phase B to reach a final volume of 1 mL for standard solutions of 50%, 75%, 100%, 125%, 150%, and 200% levels, respectively. We also prepared a quality control solution in the same way as the 100% level standard solution.

#### 2.4.2. Validation Solution

To make the validation program as close as possible to real vaccine samples, we treated mRNA, double-distilled water, DMG-PEG2K, CHOLE, DSCP, and 9001 in line with the production method of the COVID-19 mRNA vaccine to prepare spiked samples. A 100% level spiked sample contained 10 μg/mL DMG-PEG2K, 40 μg/mL cholesterol, 20 μg/mL DSCP, and 120 μg/mL 9001, and 50%, 75%, 125%, 150%, and 200% level spiked samples were prepared by adjusting the contents of the four lipids according to their levels.

#### 2.4.3. Specific Solution

The specific solutions were prepared by simulated COVID-19 mRNA vaccine without the four lipids, which contained 0.2 mg/mL COVID-19 mRNA, 0.2 mg/mL monobasic potassium phosphate, 3.07 mg/mL disodium hydrogen phosphate, and 8.8 mg/mL sodium chloride.

#### 2.4.4. Sample Preparation

Each COVID-19 mRNA vaccine sample solution was diluted 20-fold with mobile phase B for sample injection analysis.

### 2.5. Injection Sequence

Pure methanol as a blank was injected at least three times until the baseline was stable. Standard solutions were injected in order from low to high level, and then the spiked samples or COVID-19 mRNA vaccine samples were injected. The quality control solution was injected six times after the blank, and one time at the end of the sequence or after each ten injections of samples.

### 2.6. Data Analysis

The peak areas of the lipids in standard solution were normalized and integrated. The concentration of standard solution was taken as the *X*-axis, the peak area was taken as the *Y*-axis, and the standard curve of the quadratic function was generated (Y = aX^2^ + bX + C). The contents of each lipid in the spiked samples and the COVID-19 vaccine samples were calculated through regression.

### 2.7. Validation Program

Method validation was performed by two technicians in two independent laboratories. They prepared three repeats of the standard solutions, quality control solution, and spiked samples of all levels (50%, 75%, 100%, 125%, 150%, and 200%) for each of the 6 days. The system suitability requirements were as follows: the blank sample had no interference peak at the retention time of the four lipids and the *R*^2^ of the standard curve for each lipid had to not be less than 0.98. Each validation laboratory injected the spiked samples of each concentration three times; the recovery rate of each injection was then calculated, and the mean value was calculated to verify the accuracy, precision, and linearity. The average recovery rates of the six concentrations of spiked samples given through three parallel injections were calculated by two validation laboratories. Recovery rates of each concentration level within 80–120% were considered acceptable. The precision included intra-laboratory reproducibility, inter-laboratory reproducibility, and overall precision, and RSDs (relative standard deviations) not more than 10% were considered acceptable. The linearity of the HPLC-CAD method was calculated to fit the measured and theoretical concentrations of six spiked samples in two validation laboratories, and the acceptable range was that the correlation coefficient *R*^2^ was greater than or equal to 0.98. The specificity was verified by a specific solution. The acceptable criteria were that the specific sample had no interference at the retention time of any of the lipids, and the accuracy of the spiked samples at each concentration had to meet the standard requirements above.

## 3. Results

### 3.1. System Suitability

The retention times of DMG-PEG2K, CHOLE, DSCP, and 9001 were about 7.811 min, 8.126 min, 9.784 min, and 10.457 min, respectively, and the blank sample showed no interfering peaks at these time points (Figure 2). The *R*^2^ of all standard curves was greater than 0.99 throughout the validation process, indicating a good curve fit (Table 1). These data met the requirements of system suitability.

### 3.2. Accuracy

We calculated the average recovery rates of DMG-PEG2K, CHOLE, DSCP, CHOLE, and 9001 measured at each concentration three times on each of the three days in each of the two laboratories (*n* = 18). The results indicated that the average recovery of DMG-PEG2K was between 100.98% and 110.55%; the average recovery of cholesterol was between 98.10% and 108.05%; the average recovery of DSCP was between 97.34% and 102.99%; and the average recovery of 9001 was between 97.66% and 102.99%. All the recoveries of spike samples were between 90% and 110% (Figure 3).

### 3.3. Precision

The intra-laboratory reproducibility was calculated as the RSDs of the recovery rate of 18 spiked samples for each concentration level from each laboratory (Figure 4), and the inter-laboratory reproducibility was calculated as the RSDs of the recovery rate of 36 spiked samples for each concentration level and four lipids’ total contents in two validation laboratories (Figure 5). The overall precision was calculated as the RSDs of the recovery rate of all concentration levels from both laboratories (Table 2). The RSDs of all the spiked samples were below 10%.

### 3.4. Linearity

The measured values of the total content of DMG-PEG2K, CHOLE, DSCP, and 9001 in two validation laboratories were set as the *X*-axis, and the theoretical values were set as the *Y*-axis to produce a linearity curve. The curve is shown in Figure 6. The linear equations of the two laboratories were Y = 1.0029X + 2.2718 (*R*^2^ = 0.9994) and Y = 0.9945X + 2.4063 (*R*^2^ = 0.9982). This indicates that the method has good linearity.

### 3.5. Specificity

Comparison of the chromatograms of pure methanol, standard solution of level 50%, and specific solution indicated that the specific solution had no interference during the retention time of DMG-PEG2K, CHOLE, DSCP, and 9001 (Figure 7). The accuracy verification results showed that the accuracy study of the spiked samples at all concentration levels met the requirements of the standard, proving that there was no interference from the specific solution. The method can thus be considered specific.

### 3.6. Range

Based on the above validation results, the linear range of this HPLC-CAD method for the determination of the content of DMG-PEG2K, CHOLE, DSCP, and 9001 in the COVID-19 mRNA vaccines was 5–20 μg/mL, 20–80 μg/mL, 10–40 μg/mL, and 60–240 μg/mL, respectively. The accuracy and precision met the requirements in Section 2.7.

### 3.7. Sample Test

To further verify the applicability of the HPLC-CAD method, we used this method to determine the contents of four lipids in five batches of COVID-19 mRNA vaccine in two validation laboratories. The samples were prepared as in Section 2.4.3 and analyzed using the HPLC-CAD system. The content of each lipid in the sample was calculated from the standard curve and multiplied by the dilution factor of 20, which resulted in the content of each lipid in the COVID-19 mRNA vaccine. The average, RSD, and recovery rate were calculated, and the results are shown in Table 3. The RSDs were all less than 10%, and the recovery rates were all between 80% and 120%. These results indicate that this method is suitable for the determination of lipid contents in COVID-19 mRNA vaccines.

## 4. Discussion

Novel coronavirus SARS-CoV-2 has become a global health problem and has infected a significant portion of the world’s population. Over the course of the COVID-19 pandemic, people all over the world have faced significant challenges, anxiety, and stress regarding healthcare because until December 2020, there were no vaccines available for this pandemic and there is still no specific treatment. In mid-December 2020, the US Food and Drug Administration (FDA) granted emergency use authorization for Pfizer/BioNTech and Moderna COVID-19 vaccines. Many studies have analyzed and compared these two vaccines. Both vaccines can offer protection against SARS-CoV-2 infection by developing humoral immunity and possibly cellular immunity. Moreover, both vaccines are effective and offer hope for an end to the COVID-19 pandemic. The dose of the Pfizer vaccine is slightly lower (30 μg), while that of the Moderna vaccine is 100 μg. The Pfizer vaccine is approved for people aged 16 years and older, while the Moderna vaccine is approved for those 18 years and older. According to reports, the efficacy of the Pfizer vaccine is 95%, which is slightly higher than the 94.5% efficacy of the Moderna vaccine. Both vaccines can cause some adverse reactions, including pain, swelling, vomiting, nausea, fever, fatigue, headache, muscle pain, itching, chills, and joint pain at the injection site, and in rare cases, they may also cause allergic reactions. According to reports, the incidence of these adverse reactions is lower with the Pfizer/BioNTech vaccine compared to the Moderna vaccine [23,24]. However, because clinical trials of different COVID-19 mRNA vaccines were conducted under different conditions, the incidence of adverse reactions observed in the clinical trials of one vaccine cannot be directly compared with those in the clinical trials of another vaccine, nor can it reflect the actual incidence of adverse reactions observed.

Both authorized COVID-19 mRNA vaccines use modified mRNA to encode the spike protein of SARS-CoV-2, which triggers an immune response and produces neutralizing antibodies, and both use an LNP delivery system. This is a new type of vaccine that has potential advantages over traditional replicating or non-replicating viral vector vaccines because mRNA vaccines are highly effective, can be produced rapidly, and are relatively inexpensive. Compared with viral vaccines, they have good safety profiles because they are not made from actual pathogens and do not integrate with the host DNA.

Of course, both COVID-19 mRNA vaccines also have their drawbacks. Due to the instability of mRNA and the lipid nanoparticle, the quality control during the release process is difficult, and extreme cold storage is required to maintain stability during distribution [25]. The storage temperature requirement for the Pfizer vaccine is −80 °C to −60 °C (−112 °F to −76 °F), while the Moderna vaccine can be stored at a relatively higher temperature between −25 °C and −15 °C (−13 °F to −5 °F), which makes it easier to preserve. Therefore, transportation costs are also a consideration.

In the early 21st century, LNP technology for scalable production began to develop based on cationic lipid technology. It has been shown that LNP can deliver mRNA vaccines to mice, and the clinical studies have gradually attracted attention. LNPs usually consist of multiple lipid components, which are used as delivery systems to package and enhance the stability of mRNA. This can effectively avoid the degradation of mRNA outside the cell, promote its uptake by the cells, and release it into the cytoplasm. Lipid components include but are not limited to ionizable/cationic lipids, helper lipids such as neutral lipids and/or cholesterol, and lipids modified by polyethylene glycol (PEGylation). The appearance of LNPs is a milestone in the development of RNA therapy, which successfully solves the problem of protecting and delivering RNA. Currently, the two COVID-19 mRNA vaccines on the market, BNT162b2 (Comirnaty) and mRNA-1273 (Spikevax), use ionizable lipids ALC-0315 and SM-102 (Lipid H), respectively, and PEGylated lipids ALC-0159 and PEG2000-DMG [26], respectively, with a mean neutral phospholipid of dipalmitoyl phosphatidylcholine, and all use cholesterol.

As more and more COVID-19 mRNA vaccines are approved for use, the global demand for mRNA vaccines for the coronavirus presents a severe challenge for vaccine quality control. Regulatory agencies, such as the World Health Organization (WHO), the US Food and Drug Administration (FDA), the European Medicines Agency (EMA), and the Center for Drug Evaluation (CDE), have issued corresponding guidelines to standardize and guide the declaration and market application of such products. Among them, the recently released WHO document, “Evaluation of the Quality, Safety, and Efficacy of Messenger RNA Vaccines for the Prevention of Infectious Diseases: Regulatory Considerations,” is the first global regulation specifically for mRNA products that is significant for guiding national regulatory authorities to evaluate such products and clarify the regulatory requirements. The production of mRNA vaccines involves multiple biological processes and the processing of raw materials, such as large-scale in vitro transcription (IVT), mRNA processing and modification, and mRNA LNP encapsulation. During production, there is a risk of contamination and introduction of impurities, and it is necessary to study and establish multiple novel key quality control parameters and their detection methods and quality standards related to vaccine characteristics, such as identification, content, integrity, encapsulation rate, and LNP content [27]. The detection of residual template DNA, incomplete mRNA, and lipid components present difficulties for testing methods.

Currently, there is still a lack of unified testing methods and reference materials for COVID-19 mRNA vaccine quality control, and information on quality control from different research and development companies is difficult to share. How to integrate vaccine regulation, research and development, and production resources and establish unified standards and quality control systems is a key issue facing the development of the mRNA vaccine industry. Companies should design, develop, and produce vaccines based on the concept of risk management, scientifically and reasonably set quality control projects and acceptable standards, and establish simple and accurate testing methods that meet the expected purposes. Regulatory authorities should promote the uniformity of quality control standards and the application of standardized methods to provide a foundation for promoting the development of mRNA vaccines and developing safe and effective vaccines [28].

Over the past few decades, breakthroughs in nucleic acid modification and non-viral delivery vector technology have greatly promoted the development of mRNA vaccines, and the clinical application of mRNA vaccines has also validated their drug efficacy and advantages in scalable production. From a clinical perspective, long-term monitoring of adverse reactions to mRNA vaccines is still needed to respond to potential risks in a timely manner. From a drug development perspective, the current challenges that mRNA vaccines urgently need to overcome include exploring more efficient production processes to ensure their safety, effectiveness, and quality controllability, as well as finding more reasonable storage and transportation methods to improve accessibility.

There are a number of COVID-19 mRNA vaccines in the preclinical development and clinical trial stages [29]. To ensure the safety and efficacy of such vaccines, we developed and validated an efficient HPLC-CAD method to measure the lipid contents of COVID-19 mRNA LNPs. The method was validated against four kinds of lipids, namely, DMG-PEG2K, CHOLE, DSCP, and 9001 according to ICH-Q2 [30]. The results showed that the method has good accuracy, precision, linearity, and specificity. We also showed the measured results of the four lipids contents of five batches of COVID-19 mRNA vaccines. The above results indicate that the HPLC-CAD method can support the development, manufacture, and release testing of COVID-19 mRNA vaccines.

## Figures and Tables

**Figure 1 vaccines-11-00937-f001:**
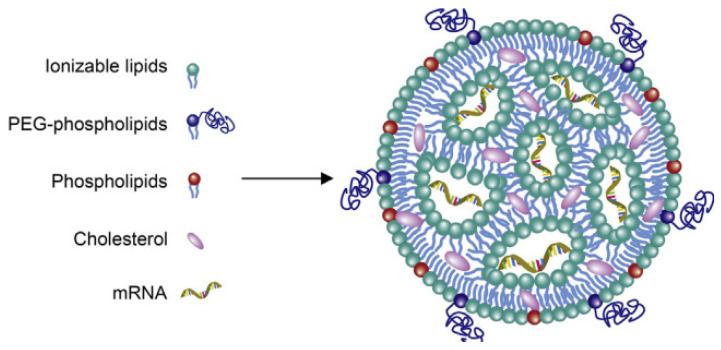
Sketch map of an LNP-encapsulated COVID-19 mRNA vaccine.

**Figure 2 vaccines-11-00937-f002:**
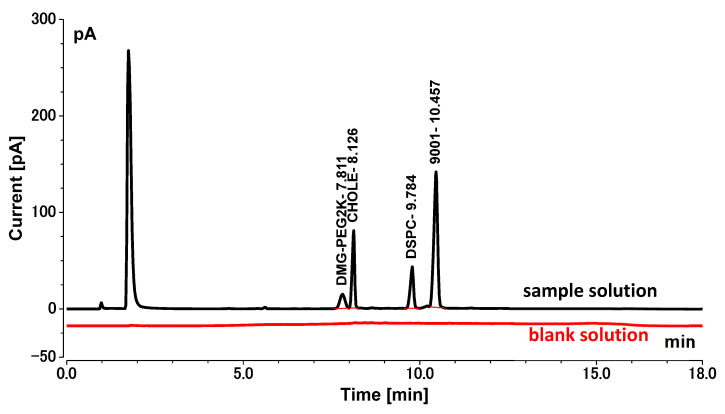
Chromatogram of blank solution and sample solution (black: sample solution; red: blank solution).

**Figure 3 vaccines-11-00937-f003:**
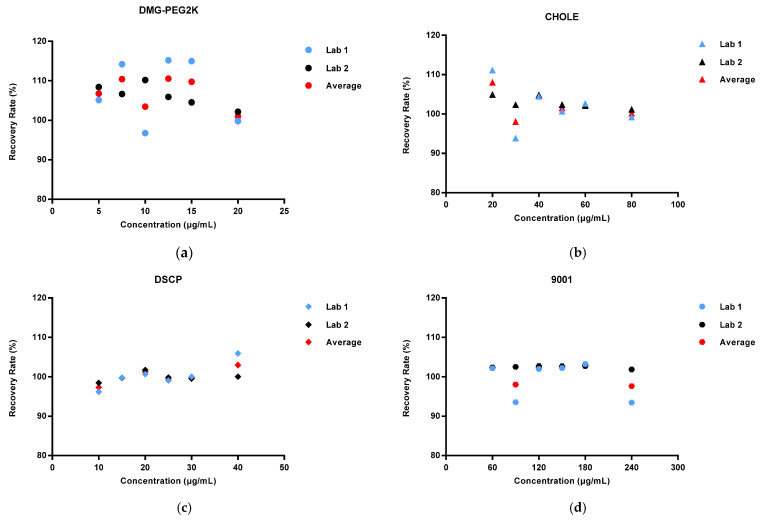
Recovery rate range in spike samples with different concentration levels (50%, 75%, 100%, 125%, 150%, and 200%) in two laboratories: (**a**) DMG-PEG2K; (**b**) CHOLE; (**c**) DSCP; (**d**) 9001.

**Figure 4 vaccines-11-00937-f004:**
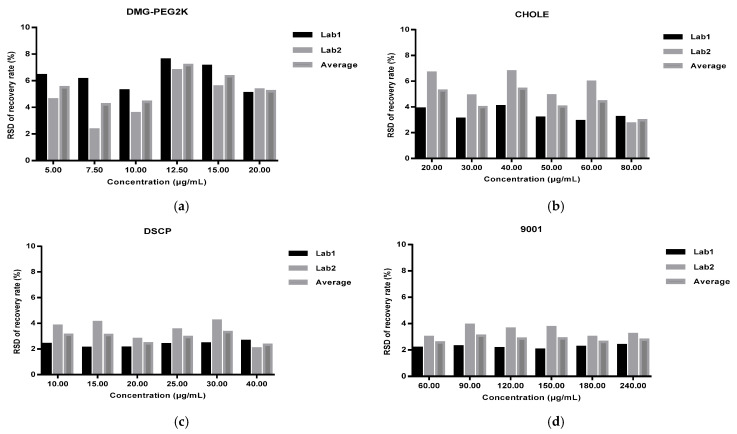
Intra-laboratory repeatability of spiked samples with different concentration levels (50%, 75%, 100%, 125%, 150%, 200%) from each laboratory: (**a**) DMG-PEG2K; (**b**) CHOLE; (**c**) DSCP; (**d**) 9001.

**Figure 5 vaccines-11-00937-f005:**
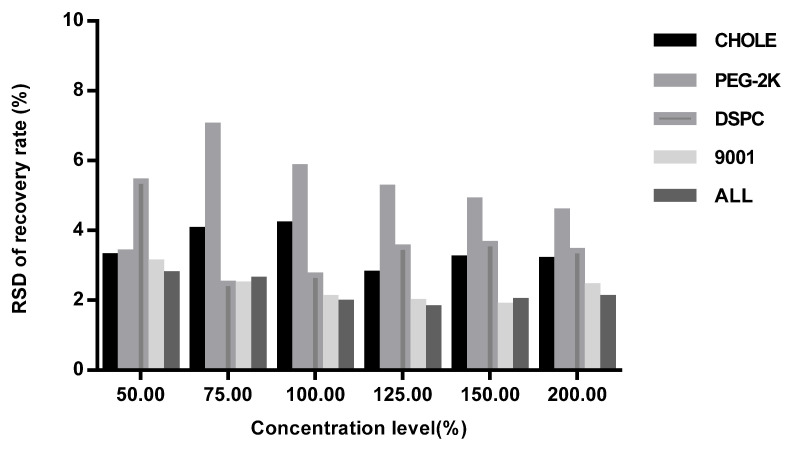
Inter-laboratory repeatability of spiked samples with different concentration levels (50%, 75%, 100%, 125%, 150%, 200%) in two validation laboratories.

**Figure 6 vaccines-11-00937-f006:**
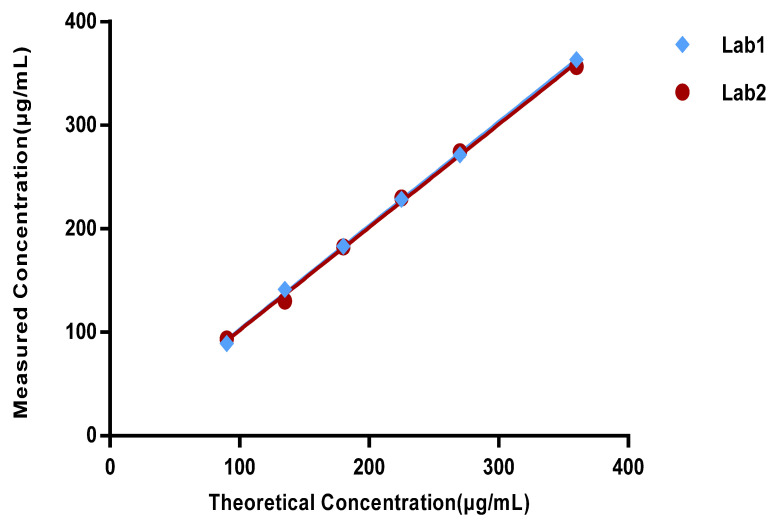
Linearity curve of two validation laboratories (total content of DMG-PEG2K, CHOLE, DSCP, and 9001).

**Figure 7 vaccines-11-00937-f007:**
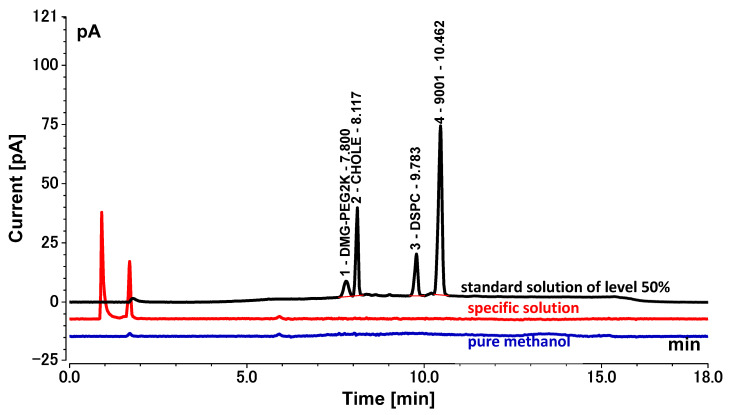
Chromatograms of pure methanol, standard solution of level 50%, and specific solution. (Black: standard solution of level 50%; blue: pure methanol; red: specific solution).

**Table 1 vaccines-11-00937-t001:** Standard curves of DMG-PEG2K, CHOLE, DSCP, and 9001.

Lipid	Standard Curve	Correlation Coefficient (*R*^2^)
DMG-PEG2K	y = −0.0026 x^2^ + 0.2779x − 0.2809	0.9930
CHOLE	y = −0.0003x^2^ + 0.1550x + 0.1933	0.9980
DSCP	y = −0.0006x^2^ + 0.1835x + 0.0268	0.9977
9001	y = −0.0001x^2^ + 0.1350x + 1.2034	0.9971

**Table 2 vaccines-11-00937-t002:** Overall precision of all concentration levels in two validation laboratories.

Validation Laboratories	RSD (%)			
	DMG-PEG2K	CHOLE	DSCP	9001
Lab1	5.29	7.16	3.17	4.89
Lab2	6.63	3.61	2.48	2.16
Overall precision	4.38	5.15	2.41	3.01

**Table 3 vaccines-11-00937-t003:** Lipids contents of COVID-19 mRNA vaccine samples.

Lipid	Theoretical Value (μg/mL)	Measured Value Lab1 (μg/mL)	Measured Value Lab2 (μg/mL)	Average (μg/mL)	SD (μg/mL)	RSD (%)	Recovery Rate (%)
DMG-PEG2K	10	9.09	9.86	9.48	0.54	5.73	94.76
CHOLE	40	39.81	41.53	40.67	1.22	2.99	101.68
DSPC	20	22.82	20.79	21.80	1.44	6.59	109.02
9001	120	129.86	121.88	125.87	5.65	4.49	104.89

## Data Availability

The qualitative data presented in this study are available on request from the corresponding author.

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
