# Peer review of "Validation of an HPLC-CAD Method for Determination of Lipid Content in LNP-Encapsulated COVID-19 mRNA Vaccines"

_vaccines, 2023, doi:10.3390/vaccines11050937_

Round 1

Reviewer 1 Report

There are no references provided that describe similar HPLC-CAD methods for lipids. The attached file indicates two; there may be more. A thorough Google Scholar literature search is needed and that HPLC-CAD information added to the Introduction. A comparison of their method to previous HPLC-DAD methods for lipids should be part of the Discussion section.

 The LOD and LOQ analytical figures of merit are missing and need to be done. Figure 6 is not complete.

Author Response

Q:There are no references provided that describe similar HPLC-CAD methods for lipids. The attached file indicates two; there may be more. A thorough Google Scholar literature search is needed and that HPLC-CAD information added to the Introduction. A comparison of their method to previous HPLC-DAD methods for lipids should be part of the Discussion section.

A:I have added relevant content in the introduction and cited references.

Q: The LOD and LOQ analytical figures of merit are missing and need to be done. Figure 6 is not complete.

A:According to the requirements of ICHQ2 and Chinese Pharmacopoeia for methodology verification of content determination items, the verification content related to LOD and LOQ has not been carried out in this verification, there is only range, which will be gradually improved in the future according to actual needs. Figure 6 is update.

Reviewer 2 Report

The manuscript presents the HPLC-CAD method for the determination of lipid content in LNP-encapsulated COVID-19 mRNA vaccines.

The overall manuscript is interesting, but it has to be carefully corrected. 

Part 3.1 A chromatogram should be provided. 

Line 174 It looks like the RSD is below 9% or 8%. However, it is difficult to read the figure.

Fig. 2, Fig. 3, and Fig. 6 are unreadable.

Fig. 5 Detailed description of the figure is necessary. What kind of substance is presented?

Author Response

Q: Part 3.1 A chromatogram should be provided. 

A: I have provide a chromatogram.

Q :Line 174 It looks like the RSD is below 9% or 8%. However, it is difficult to read the figure.

A: Methodological verification requires RSD less than 10%, and Figure has been updated.

Q: Fig. 2, Fig. 3, and Fig. 6 are unreadable.

A: Figures have been updated

Q: Fig. 5 Detailed description of the figure is necessary. What kind of substance is presented?

A: It is the linearity curve of total content of DMG-PEG2K, CHOLE, DSCP, and 9001. X-axis means theoretical concentration and Y-axis means measured concentrion.And I have added description on the Figure.

Reviewer 3 Report

Comments to: ‘Validation of an HPLC-CAD method for determination of lipid 2 content in LNP-encapsulated COVID-19 mRNA vaccines’ Yu et al., 2023

This study describes the development and validation of an analytical procedure to quantify the lipid components of LNP-mRNA vaccines.

Major questions:

·        I leave it to the editors to decide whether this topic (analytical method validation) falls under the ‘subject area’ of the journal.

·        When reading this text I wonder what the added value of this text is compared to the publication of Kinsey et al, 2022 Electrophoresis 2022, 43, 1091–1100.

Other comments:

Line 51-53: May I suggest that the authors add some text explaining how they came to use this analytical configuration (compared to other protocols/techniques described in the literature for lipid analysis).

The mRNA must be fully separated from the (cationic) lipid in the analytical runs. Did the authors confirm this?

The ‘specific solution’ should have contained sucrose as excipient.

In the ‘validation program’ section: what is the ground for the acceptability ranges mentioned? Are regulatory authorities going to accept these ranges?

Does this analytical set up pick up degradants/impurities?

Figure 2: green and red dots: difficult to read by color blind colleagues. So, please change the color code.

Figure 6 is unreadable.

Line 225: Spikevax is not the most widely used COVID-19 mRNA vaccine. That is the Pfizer-BioNTech Comirnaty®.

Author Response

  • Q:  I leave it to the editors to decide whether this topic (analytical method validation) falls under the ‘subject area’ of the journal.
  • When reading this text I wonder what the added value of this text is compared to the publication of Kinsey et al, 2022 Electrophoresis 2022, 43, 1091–1100.

A:Compared with this paper, our study has the following innovations: First, our stduy is a joint verification of two laboratories, and the verification content and scheme are more complete and the data are more accurate. Secondly, the CoVID-19 mRNA vaccine in our study does not require pretreatment, so the method is much simpler.

Q: Line 51-53: May I suggest that the authors add some text explaining how they came to use this analytical configuration (compared to other protocols/techniques described in the literature for lipid analysis).

A: Relevant content is added in the introduction part .  

Q: The mRNA must be fully separated from the (cationic) lipid in the analytical runs. Did the authors confirm this?

A: Each COVID-19 mRNA vaccine sample solution was diluted 20-fold with mobile phase B for sample injection analysis, phase B was an organic eluent containing 0.01 M triethylamine acetate in ethyl alcohol. So mRNA and (cationic) lipid were seperated during analysis process. And on the chromatogram map, the retention time of mRNA is very different from (cationic) lipid.

Q: The ‘specific solution’ should have contained sucrose as excipient.

A: The COVID-19 mRNA vaccine does not contain sucrose, so the 'specific solution' in the methodological validation does not require the addition of sucrose.

Q: In the ‘validation program’ section: what is the ground for the acceptability ranges mentioned? Are regulatory authorities going to accept these ranges?

A:The Validation program is based on the ICHQ2 guidelines and the Chinese Pharmacopoeia and is generally recognized by the regulatory authorities.

Q: Does this analytical set up pick up degradants/impurities?

A: At present, this method is only used for the release quality control of mRNA vaccine and does not achieve significant performance to achieve degradants/impurities, and then it is added according to needs.

Q: Figure 2: green and red dots: difficult to read by color blind colleagues. So, please change the color code.

A:  I have changed the color.

Q: Figure 6 is unreadable.

A: The figure is update.

Q: Line 225: Spikevax is not the most widely used COVID-19 mRNA vaccine. That is the Pfizer-BioNTech Comirnaty®.

A: I have changed.

Round 2

Reviewer 1 Report

LOD and LOQ can easily be determined based respectively as 3 or 10 times the standard deviation of the calibration curve intercept (determined by LINEST in Excel) divided by the slope.

Author Response

Firstly, simple estimation of QL without validation is not accepted.

As stated in ICH Q2(R1), LOQ could be determined by several ways, including signal to noise to ratio of 10 (Section 7.2). However, the limit should be subsequently validated by the analysis of a suitable number of samples known to be near or prepared at the quantitation limit as described in section of 7.4.

Furthermore, the new version of ICH Q2(R2) also stated that quantitation limit (QL) can be estimated using different approaches (Section 4.2.2) also including signal to noise to ratio of 10 (Section 4.2.2), and made it very clear that if the QL was estimated, the limit should be subsequently validated by the analysis of a suitable number of samples known to be near or at the QL.

So according to the ICH guideline, the QL should be meet certain criteria of accuracy and precision.

Secondly, QL and DL validation are not needed in the content method such as described in the manuscript as stated in the table at Page 3 of ICH Q2(R1). Furthermore, ICH Q2(R2) made it very clear that in cases where the QL is well below the reporting limit, this confirmatory validation can be omitted with justification (Section 4.2.2.4). The lower limit of the standard curve is well below the reporting limit of the method, so the validation of the lower limit of the standard curve fit for the purpose of the method.

As stated above, we do not think the validation of QL is needed, neither DL. 

Reviewer 2 Report

The manuscript was corrected. The Authors responded to all comments. 

I can recommend the manuscript for publication.

Author Response

Thanks very much for your recommendation. And I have polished the English.

Reviewer 3 Report

figures 2 and 7 are still unreadable. 

line 278: The mRNA dose of the Moderna vaccine (first series) for adults is typically 100 microgram..... as far as I know.

line 315: Onpattro is not a vaccine and doesn't contain mRNA but siRNA.

Author Response

I have modified the manuscript according to your suggestion and polished the English by professional institute and get the certificate.